# Risk of deep venous thrombosis associated with peripherally inserted central catheter: A retrospective cohort study of 11.588 catheters in Brazil

**Telma Christina do Campo Silva**[1]*, **Luciene Muniz Braga**[2], **Jose Mauro Vieira Junior**[3]

**1** Department Nursing Specialized in Vascular Access, RN, MD. Hospital Sírio Libanês, São Paulo, São Paulo, Brazil, **2** Adjunct Professor at the Federal University of Viçosa, RN, PhD. Viçosa, Minas Gerais, Brazil, **3** Physician at the Dialysis Center, MD, PhD. Hospital Sírio Libanês, Nephrologist and Professor at Hospital das Clínicas of the University of São Paulo, São Paulo, São Paulo, Brazil

* telmachriscampo@gmail.com

**Data Availability Statement:** All relevant data is in the manuscript and its supporting information files.

**Funding:** The author didn't receive specific funding for this work.

## Abstract

### Introduction

Deep Venous Thrombosis (DVT) due to Peripherally Inserted Central Catheter (PICC) is one of the most threatening complications after device insertion.

### Objective

To assess the rate of PICC-associated DVT and analyze the risk factors associated with this event in cancer and critically ill patients.

### Methods

We conducted a descriptive, retrospective cohort study with 11,588 PICCs from December 2014 to December 2019. Patients $\geq$ 18 years receiving a PICC were included. Pre-and post-puncture variables were collected and a logistic regression was used to identify the independent factors associated with the risk of DVT.

### Results

The DVT prevalence was 1.8% (n = 213). The median length of PICC use was 15.3 days. The median age was 75 years (18; 107) and 52% were men, 53.5% were critically ill and 29.1% oncological patients. The most common indications for PICC's were intravenous anti-biotics (79.1%). Notably, 91.5% of PICC showed a catheter-to-vein ratio of no more than 33%. The tip location method with intracavitary electrocardiogram was used in 43%. Most catheters (67.9%) were electively removed at the end of intravenous therapy. After adjusting for cancer profile ou chemotherapy, regression anaysis revealed that age (OR 1.011; 95% CI 1.002–1.020), previous DVT (OR 1.96; 95% CI 1.12–3.44) and obstruction of the device (OR 1.60; 95% CI 1.05–2.42) were independent factors associated with PICC-associated

**Competing interests:** The authors declared that there are no conflicting interests.

DVT, whereas the use of an anticoagulant regimen was a protective variable (OR 0.73; 95% CI 0.54–0.99).

## Conclusion

PICC is a safe and suitable intravenous device for medium and long-term therapy, with low rates of DVT even in a cohort of critically ill and cancer patients.

## Introduction

There have been several advancements available for patients needing prolonged intravenous therapy, from which the peripherally inserted central catheter (PICC) is highlighted for having advantages over other devices, including a reduced risk of complications related to insertion, such as pneumothorax, easy maintenance and dehospitalization of patients undergoing antibiotic therapy or chemotherapy [1–4].

The main indications for PICC are intravenous therapy for $\geq$14 days, intravenous therapy with incompatibility for peripheral venous access, critically-ill patients with bleeding disorders, the continuous use of vesicant infusions, such as parenteral nutrition or irritant solutions, cyclic chemotherapy treatment, and patients under palliative care [5].

In order to obtain good results with PICC use, the Infusion Nurses Society (INS) guidelines recommend vessel occupancy $\leq$ 45%, and insertion of the catheter in ideal sites using the zone insertion method (ZIM). Technologies related to PICC insertion, device characteristics, employing the best practices, as the use of ultrasound to ensure puncture in the ideal zone, calculation of vessel occupancy according to the catheter used, and the advent of vascular access teams have improved the results of these catheters [1,6–9].

However, PICC use is still associated with risks such as infection and deep venous thrombosis (DVT). Catheter-related DVT is a serious vascular access complication that can lead to the development of pulmonary thromboembolism (PTE), infection, access dysfunction, and post-thrombotic syndrome. These complications can not only interrupt the treatment, but also in addition increase costs, morbidity, and mortality [8,10].

The worldwide use of PICC is growing and services experiences with its employement have been described. Here, our objective is to describe the actual rate and risk factors associated with symptomatic DVT in PICC patients, in a tertiary brazilian hospital, particularly in a scenario of cancer and critically-ill patients.

## Materials and methods

### Study design and recruitment criteria

This study was a descriptive and retrospective cohort study involving all adult patients at the Sírio Libanês Hospital (HSL), a tertiary, high acuity, JCI-accredited hospital in São Paulo, Brazil, undergoing PICC insertion from December 2014 to December 2019.

The Research Electronic Data Capture (REDCap) was used for data collection. The patients with PICC-associated DVT diagnosis were compared to the rest of the database. Inpatients and outpatients with PICCs dwell time $\geq$3 days were included [11]. (**Fig 1**)

Venous Doppler was performed only in patients with symptoms, such as pain, increased arm circumference, pain in the axillary region, and edema of the extremities. An active ultrasound-armed surveillance search for PICC-associated DVT was not routinely performed.

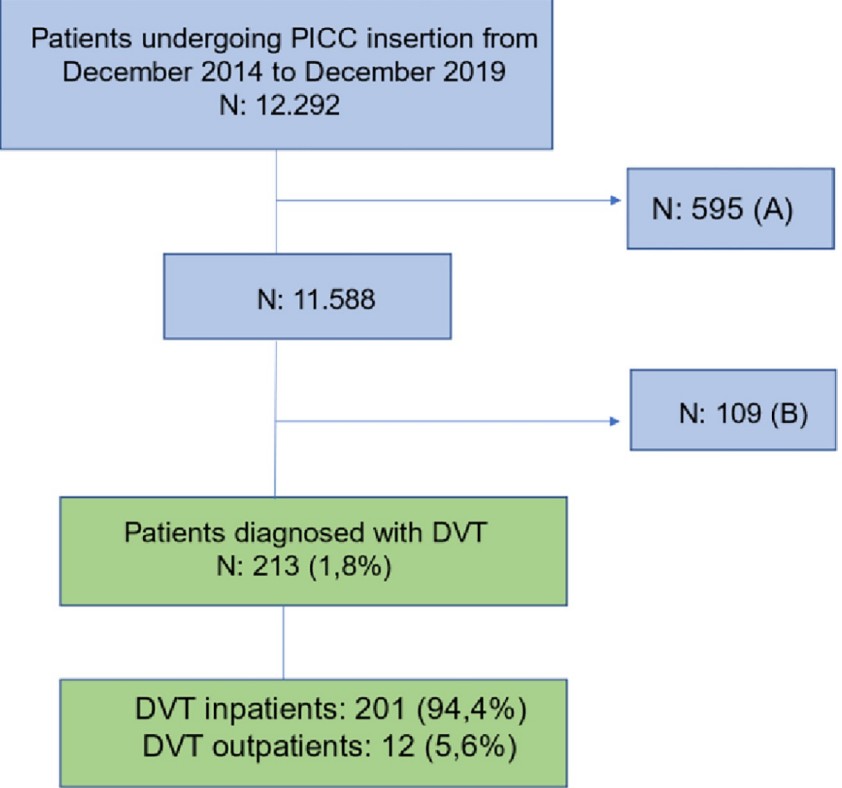

Fig 1. Data collection flowchart.

## Inclusion and exclusion criteria

Inclusion Criteria: Adult in and outpatients, undergoing PICC insertion using the modified Seldinger technique by the exclusive dedicated vascular access nurses team at the HSL.

Exclusion criteria: Patients with a catheterization duration of $\leq$ 3 days; $<$ 18 years-old.

## Study variables

The variables analyzed were age, sex, reason and site of admission, severity of the disease, oncological or surgical patient profile, previous DVT, use of oral or parenteral anticoagulants, number of vein puncture attempts, punctured limb, puncture area, chosen venous access, vessel occupancy, type of PICC, catheter tip location at the end of the procedure.

## Data collection and quality assurance

All patients used catheters from a single brand, chosen by the institution (Becton Dickinson—BD®). The device materials were polyurethane and silicone. Open-tip and valved catheters (anti-reflux, Groshong®) were used.

The ideal zone in the green area was the first choice for PICC insertion. All catheters were inserted by a team of ten fully trainned and dedicated nurses, using the ultrasound-guided at the bedside or in the operating room, with maximum barrier protection [7].

In relation to TIP location assessing method, the first choice as October 2016 was intracavitary ECG and in cases of impossibility of using this technology, the navigation resource with confirmation through chest x-ray or scopy was used.

In cases where the x-ray was used, the tip location was defined according Sweet-spot. The technique was created to reduce complications caused by vascular access. A rectangular template is superimposed on a frontal radiograph whose internal margins are acceptable for the catheter tip position. It has fixed portions called A, B and C. Measurement of catheter occupancy in relation to vein size was collected directly by Site-Rite 5® or Site-Rite 8® ultrasound equipment [6,12].

The current Infusion Nurses Society guidelines, recommend the use of PICC's with a catheter-to-vein ratio of no more than 45% prior to insertion of a vascular access device in the upper extremity, and we follow this guideline [6].

The precautions recommended for PICC maintenance bundle were performed first by dressing with gauze and sterile transparent film, and eventually changed to a medicated transparent film after 24 h, and a fixation device without suture (Statlock®) was used in all devices in order to avoid dislodgement.

## Statistical methods

The continuous variables were analyzed for distribution parametric versus nonparametric and were presented as median and interquartil interval (IQ) for nonparametric and mean ± standard deviation (SD) for continuous parametric variables. Groups (PICC-related DVT and the controls) were compared using the Student's t-test or the Mann-Whitney test, when applicable. The categorical variables were described using absolute and relative frequencies; additionally, the difference was verified with the chi-square or exact tests (Fisher's exact test or likelihood ratio test).

The odds ratio (OR) of each variable evaluated were estimated for the occurrence of DVT with the respective 95% confidence intervals (CI) using unadjusted and adjusted logistic regression. A joint model was built to explain the occurrence of DVT using multiple logistic regression. A significance level of 5% was set for the tests.

## Ethics statement

The data were collected after research project approval by the HSL Institutional Review Board (IRB) (approval number CAAE 99329118.6.0000.5461). The requirement for informed consent from participants was waived owing to the retrospective characteristics of the study, based on data collection in the REDCap system.

## Results

From December 2014 to December 2019, a total of 12,292 patients underwent PICC insertion. Of these, 595 patients were excluded due to age <18 years and 109 patients were excluded with PICCs dwell time ≤ 3 days. In total, 11,588 procedures were included. Regarding the type of care, 11,009 patients (95%) were inpatients whereas only 574 (5%) were outpatients.

The overall DVT prevalence rate in the period considered for the study was 1.8% (n = 213), from which 201 occurred in inpatients (94.4%) and 12 (5.6%) in outpatients.

## General characteristics of the cohort

The population consisted of 52% male (n = 6,021). The mean age was 70.6 ± 18.2 years. The mean duration of PICC placement was 15.3 ± 19.3 days. Among the participants, 53.5% (n = 6,156) were critically ill, and 29.1% (n = 3,334) were cancer patients. Concerning

**Table 1. General characteristics (n = 11,588).**

| Category/Variable | n | % |
|---|---|---|
| **Sex** | | |
| Men | 6021 | 52 |
| Women | 5567 | 48 |
| **Age (years)** | | |
| Mean ± SD* | 70.6 ± 18.2 | |
| Median (min, max) | 75 (18, 107) | |
| **PICC usage time (days)** | | |
| Mean ± SD* | 15.3 ± 19.3 | |
| Median (min, max) | 11 (0, 395) | |
| **Patient characterization** | | |
| Critical | 6156 | 53.5 |
| Oncological | 3334 | 29.1 |
| Surgical | 3080 | 26.9 |
| Active cancer | 2375 | 72.7 |
| Previous DVT* | 422 | 3.8 |
| **Indication for PICC insertion** | | |
| Antibiotic | 9149 | 79.1 |
| Irritating or vesicant drugs (pH <5 or >9) | 5149 | 44.5 |
| Damaged peripheral venous system | 4455 | 38.6 |
| Hypertonic solutions | 3077 | 26.6 |
| Vasoactive drugs | 1761 | 15.2 |
| Chemotherapy | 998 | 8.6 |
| Parenteral nutrition | 542 | 4.7 |
| Transfusion | 270 | 2.4 |
| **Anticoagulant therapy** | 4912 | 43.3 |
| Low molecular weight heparin | 2731 | 56.1 |
| Unfractionated heparin | 1624 | 33.3 |
| Rivaroxaban | 271 | 5.6 |
| Warfarin | 125 | 2.5 |
| Apixaban | 103 | 2.1 |
| Dabigratan etexilate | 13 | 0.3 |
| Fondaparinux sodium | 3 | 0.1 |

*DVT—Deep Venous Thrombosis; SD—Standard Deviation.

intravenous therapy, most patients underwent PICC insertion for antibiotic administration 79.1%, (n = 9,149). As for anticoagulant therapy, 43.3% (n = 4.912) of the patients used anticoagulants during the use of PICC for either prophylactic or therapeutic indications. Low molecular weight heparin was used in 56.1% (n = 2,731) (**Table 1**).

## Provider, device and insertion characteristics

Table 2 shows that 51.2% (n = 5,914) of the PICC were inserted in the right upper limb (RUL). Notably, 88.4% (n = 10,176) of insertions were performed in the first puncture attempt. The basilic vein was the first choice in 61% (n = 7,011), 78% (n = 9,001) being inserted in the green zone. Regarding vessel occupancy, in the period that *Site-Rite 5* was used, 94.6% (n = 4,280) procedures had a vessel occupancy ≤ 33% (the *Site-Rite 5* does not have the resource for

**Table 2. Device characteristics.**

| Variables | N | % |
|---|---|---|
| **Member** | | |
| RUL* | 5914 | 51.2 |
| LUL* | 5633 | 48.8 |
| | | |
| **Missing data: 41** | | |
| **Number of puncture attempts** | | |
| 1 | 10176 | 88.4 |
| 2 | 870 | 7.6 |
| 3 | 324 | 2.8 |
| 4 | 136 | 1.2 |
| **Missing data: 82** | | |
| **Chosen venous access** | | |
| Basilic | 7011 | 61 |
| Brachial | 4362 | 38 |
| Cephalic | 110 | 1 |
| Middle cubital | 8 | 0.1 |
| Saphena | 1 | 0 |
| Axillary | 1 | 0 |
| **Missing data: 95** | | |
| **Venipuncture area** | | |
| Green zone | 9045 | 78 |
| Ideal zone | 2192 | 19 |
| Yellow zone | 267 | 2.3 |
| Red zone | 84 | 0.7 |
| **Equipment used** | | |
| *Site-Rite 5* | 4719 | 40.7 |
| *Site-Rite 8* | 6869 | 59.3 |
| **Site-Rite 5 vessel occupancy** | | |
| ≤33% | 4280 | 94.6 |
| ≥33% | 246 | 5.4 |
| **Site-Rite 8 vessel occupancy** | | |
| **1–10%** | 1495 | 22.6 |
| 11–20% | 3630 | 54.8 |
| 21–30% | 1197 | 18.1 |
| 31–40% | 295 | 4.5 |
| 41–50% | 10 | 0.2 |
| >51% Missing data: 434 | 1 | 0 |
| **Type of catheter** | | |
| 4 Fr* ML* open-tip polyurethane | **1965** | **17** |
| 5 Fr ML open-tip polyurethane | 1194 | 10 |
| 5 Fr DL* open-tip polyurethane | **5015** | **43.3** |

(*Continued*)

**Table 2.** (Continued)

| Variables | N | % |
|---|---|---|
| 5 Fr TL* polyurethane | 3 | 0 |
| 6 Fr DL open-tip polyurethane | **2995** | **25.9** |
| 6 Fr TL open-tip polyurethane | 409 | 3.2 |
| 4 Fr ML Groshong valve silicone | 7 | 0.1 |
| Radiography/Navigation/Scopy (Proper positioning) | **5759** | **57** |
| **Zone A (atrium-Cava junction)** | 3851 | 66.9 |
| Zone B (superior vena cava) | 1379 | 23.9 |
| Zone C (brachycephalic vein) | 529 | 9.2 |
| Intracavitary ECG* | **4976** | **43** |
| **Missing data: 853** | | |

*RUL–a; *LUL–Left upper limb; *Fr–French; *ML—Mono lumen; *DL—Double lumen; *TL–Triple lumen; *ECG–electrocardiogram.

measuring the vessel in a three dimensional way), and with *Site-Rite 8* usage, most procedures 54.8% (n = 3,630) had a vessel occupancy of 11–20%. The first catheter choice was the 5 Fr DL (double-lumen French) open-tip polyurethane in 43.3% (n = 5,015), and 70% of all devices were 5Fr or smaller (**Table 2**).

About tip location, radiography/navigation scopy was used in 57% (n = 5,759) of the procedures, and in these procedures PICC tip final position was located in zone A in 66.9% (n = 3,851). Proper tip location was verified by intracavitary ECG in 43% of the procedures (n = 4,976).

## Device complications and outcomes

Symptomatic DVT was the most prevalent complication, ocurring in 1.8% (n = 213) of PICCs. In 80% of cases associated with PICC, DVT occurred before 20 days of PICC placement. Confirmed CLABSI occurred in 0.9% (n = 107) patients, but suspected CLABSI was noted in 14.1% (n = 1,183) patients, which led to cathether removal. In relation to minor complications, reversible catheter oclusion occurred in 8.7% (n = 1011) and acidental dislodgment in 2.6% (n = 218). The most common reasons for removing the catheter was the conclusion of infusion therapy, 67.9% (n = 5711). Eleven percent of patients died during follow-up, due to their underlying conditions.

The results of the unadjusted analyses of the pre-puncture characteristics. Only the indication for use of chemotherapy, the characterization of the cancer patient profile, and previous DVT showed a statistically significant association with the occurrence of PICC-associated DVT (p = 0.002, p = 0.041, and p = 0.024, respectively) **Table 3**.

The next table shows the results of unadjusted analyses of the post-puncture characteristics. We found that the application of a high pressure saline flush in cases in which PICC repositioning was necessary statistically associated with a decrease in the occurrence of PICC-associated DVT (p = 0.030). Additionally, reversible catheter oclusion as an adverse event showed a statistically significant association with the occurrence of PICC-related thrombosis (p = 0.039). None of the technical aspects of the insertion, such as choice of the site, type and size of the cathether, vessel occupancy ratio or cathether tip location had association with the risk of PICC-related thrombosis (**Table 4**).

**Table 3. Description of DVT occurrence by pre-puncture characteristics and unadjusted analysis results.**

| Variable | Thrombosis (PICC) | | OR | 95% CI | | p |
|---|---|---|---|---|---|---|
| | **No** | **Yes** | | **Inferior** | **Superior** | |
| **Sex, n (%)** | | | | | | 0.230 |
| Men | 5919 (98.3) | 102 (1.7) | 1.00 | | | |
| Women | 5456 (98) | 111 (2) | 1.18 | 0.90 | 1.55 | |
| **Age (years)** | | | 1.005 | 0.997 | 1.013 | 0.215** |
| Mean ± SD | 70.6 ± 18.2 | 72.1 ± 18.2 | | | | |
| Median (min, max) | 75 (18, 107) | 77 (21, 102) | | | | |
| **Antibiotic, n (%)** | | | | | | 0.351 |
| No | 2367 (97.9) | 50 (2.1) | 1.00 | | | |
| Yes | 8986 (98.2) | 163 (1.8) | 0.86 | 0.62 | 1.18 | |
| **DVA, n (%)** | | | | | | 0.146 |
| No | 9624 (98.2) | 173 (1.8) | 1.00 | | | |
| Yes | 1721 (97.7) | 40 (2.3) | 1.29 | 0.91 | 1.83 | |
| **Irritating or vesicant drugs, n (%)** | | | | | | 0.322 |
| No | 6299 (98.3) | 111 (1.7) | 1.00 | | | |
| Yes | 5047 (98) | 102 (2) | 1.15 | 0.87 | 1.50 | |
| **Hypertonic solutions, n (%)** | | | | | | 0.094 |
| No | 8316 (98) | 167 (2) | 1.00 | | | |
| Yes | 3031 (98.5) | 46 (1.5) | 0.76 | 0.54 | 1.05 | |
| **Parenteral nutrition, n (%)** | | | | | | 0.741 |
| No | 10811 (98.2) | 202 (1.8) | 1.00 | | | |
| Yes | 531 (98) | 11 (2) | 1.11 | 0.60 | 2.05 | |
| **Damaged peripheral venous system, n (%)** | | | | | | 0.917 |
| No | 6970 (98.2) | 131 (1.8) | 1.00 | | | |
| Yes | 4374 (98.2) | 81 (1.8) | 0.99 | 0.75 | 1.30 | |
| **Chemotherapy, n (%)** | | | | | | **0.002** |
| No | 10373 (98.3) | 182 (1.7) | 1.00 | | | |
| Yes | 967 (96.9) | 31 (3.1) | 1.83 | 1.24 | 2.69 | |
| **Transfusion, n (%)** | | | | | | 0.643* |
| No | 10938 (98.2) | 205 (1.8) | 1.00 | | | |
| Yes | 264 (97.8) | 6 (2.2) | 1.21 | 0.53 | 2.76 | |
| **Critical, n (%)** | | | | | | 0.741 |
| No | 5254 (98.1) | 101 (1.9) | 1.00 | | | |
| Yes | 6045 (98.2) | 111 (1.8) | 0.96 | 0.73 | 1.25 | |
| **Oncological, n (%)** | | | | | | **0.041** |
| No | 7995 (98.3) | 137 (1.7) | 1.00 | | | |
| Yes | 3259 (97.8) | 75 (2.2) | 1.34 | 1.01 | 1.79 | |
| **Surgical, n (%)** | | | | | | 0.750 |
| No | 8230 (98.2) | 153 (1.8) | 1.00 | | | |
| Yes | 3021 (98.1) | 59 (1.9) | 1.05 | 0.78 | 1.42 | |
| **Previous thrombosis, n (%)** | | | | | | **0.024** |
| No | 10571 (98.2) | 194 (1.8) | 1.00 | | | |
| Yes | 408 (96.7) | 14 (3.3) | 1.87 | 1.08 | 3.25 | |
| **Active cancer** | | | | | | 0.670 |
| No | 873 (97.7) | 21 (2.3) | 1.00 | | | |
| Yes | 2325 (97.9) | 50 (2.1) | 0.89 | 0.53 | 1.50 | |
| **Anticoagulants, n (%)** | | | | | | 0.069 |

*(Continued)*

**Table 3.** (Continued)

| Variable | Thrombosis (PICC) | | OR | 95% CI | | p |
|---|---|---|---|---|---|---|
| | **No** | **Yes** | | **Inferior** | **Superior** | |
| No | 6299 (98) | 129 (2) | 1.00 | | | |
| Yes | 4836 (98.5) | 76 (1.5) | 0.77 | 0.58 | 1.02 | |
| **Number of puncture attempts, n (%)** | | | | | | 0.490# |
| 1 | 9994 (98.2) | 182 (1.8) | 1.00 | | | |
| 2 | 849 (97.6) | 21 (2.4) | 1.36 | 0.86 | 2.15 | |
| 3 | 320 (98.8) | 4 (1.2) | 0.69 | 0.25 | 1.86 | |
| 4 | 133 (97.8) | 3 (2.2) | 1.24 | 0.39 | 3.93 | |
| **Punctured limb, n (%)** | | | | | | 0.183 |
| LUL | 5520 (98) | 113 (2) | 1.00 | | | |
| RUL/RLL | 5817 (98.3) | 99 (1.7) | 0.83 | 0.63 | 1.09 | |

Chi-square test

# Likelihood ratio test

** Student's t-test.

DVA: Vasoactive drug.

The result of the model adjusted for patients using chemotherapy and cancer patients. Overall, the chance of PICC-related DVT increased with each year of increasing age (OR, 1.011; 95% CI, 1.002–1.020), patients with previous DVT were more likely to have PICC-related DVT (OR, 1.96; 95% CI, 1.12–3.44), the use of anticoagulants reduced the chance of PICC-related DVT (OR, 0.73; 95% CI, 0.54–0.99), and the occurrence of obstruction increased the chance of PICC-related thrombosis (OR, 1.60; 95% CI, 1.05–2.42) (**Table 5**).

## PICC-related DVT

In the period determined for the study, 11,588 procedures were included. Symptomatic PICC-related DVT was identified in 1.8% (n = 213) of the cases. *Kaplan-Meier* analysis demonstrated that approximately 80% of the cases of PICC-associated DVT occurred within 20 days of catheter insertion (**Fig 2**).

## Discussion

This is a descriptive and retrospective cohort study with patients conducted in Brazil, with a robust sample of adult patients (11,588) with PICC inserted by a dedicated team of nurses with expertise in vascular access. We provide insights about indications of PICC use, insertion techniques and outcomes.

The rate of PICC-associated DVT was 1.8% in this study. We understand that this is a very low rate, considering the characteristics of the population studied. Previously, a systematic review and meta-analysis found a DVT rate of 1–3% in non-cancer patients and rate of 5–6% in cancer patients. Balsonaro et al, in a meta-analysis that included only studies in which catheter insertion had been performed according to good clinical practices, PICC-related DVT was 2.4% in non-cancer patients, 2.2% in cancer patients and 5.9% in hematologic patients. Recently, Bahl and colleagues, in another systematic review, found DVT rates ranging from 0.9 to 10%, depending on the catheter diameter, and likewise they did not find higher risk in oncological patients. Our results are in agreement with that study, since for PICC devices not larger than 5 Fr, the DVT rate stays very low. In another multicenter study, data from 16

**Table 4. Description of DVT occurrence by post-puncture characteristics and unadjusted analysis results.**

| Variable | Thrombosis (PICC) | | OR | 95% CI | | p |
|---|---|---|---|---|---|---|
| | No | Yes | | Inferior | Superior | |
| **Vessel occupancy 33%, n (%)** | | | | | | 0.771 |
| No | 10526 (98.2) | 195 (1.8) | 1.00 | | | |
| Yes | 542 (98) | 11 (2) | 1.10 | 0.59 | 2.02 | |
| **Venipuncture area, n (%)** | | | | | | 0.112# |
| Green area | 8828 (98.1) | 173 (1.9) | 1.14 | 0.80 | 1.63 | |
| Yellow area | 265 (99.3) | 2 (0.7) | 0.44 | 0.11 | 1.83 | |
| Red area | 84 (100) | 0 (0) | & | | | |
| Ideal zone | 2155 (98.3) | 37 (1.7) | 1.00 | | | |
| **Chosen venous access, n (%)** | | | | | | 0.329# |
| Basilic | 6874 (98) | 137 (2) | 1.00 | | | |
| Brachial | 4290 (98.3) | 72 (1.7) | 0.84 | 0.63 | 1.12 | |
| Others | 119 (99.2) | 1 (0.8) | 0.42 | 0.06 | 3.04 | |
| **Type of catheter, n (%)** | | | | | | 0.161# |
| 3 Fr Mono | 65 (97) | 2 (3) | 1.00 | | | |
| 4 Fr Mono | 1949 (98.8) | 23 (1.2) | 0.38 | 0.09 | 1.66 | |
| 5 Fr Mono | 1132 (98.2) | 21 (1.8) | 0.60 | 0.14 | 2.63 | |
| 4 Fr Double/5 Fr Double | 4925 (98.1) | 95 (1.9) | 0.63 | 0.15 | 2.60 | |
| 6 Fr Double | 2932 (97.9) | 63 (2.1) | 0.70 | 0.17 | 2.92 | |
| 5 Fr Triple/6 Fr Triple | 363 (97.6) | 9 (2.4) | 0.81 | 0.17 | 3.81 | |
| **Central tip location, n (%)** | | | | | | 0.128# |
| No | 849 (97) | 26 (3) | 1.00 | | | |
| Yes | 5733 (98.1) | 110 (1.9) | 0.63 | 0.41 | 0.97 | |
| Yes, with fold in the tip | 60 (98.4) | 1 (1.6) | 0.54 | 0.07 | 4.08 | |
| **Vena cava, n (%)** | | | | | | 0.429# |
| Superior vena cava | 5668 (98.1) | 109 (1.9) | 1.00 | | | |
| Anomalous vena cava | 31 (100) | 0 (0) | & | | | |
| Inferior vena cava | 14 (100) | 0 (0) | & | | | |
| **Vena cava zone, n (%)** | | | | | | 0.477 |
| Zone A | 3784 (98.3) | 67 (1.7) | 1.00 | | | |
| Zone B | 1349 (97.8) | 30 (2.2) | 1.26 | 0.81 | 1.94 | |
| Zone C | 517 (97.7) | 12 (2.3) | 1.31 | 0.70 | 2.44 | |
| **Other veins, n (%)** | | | | | | 0.160# |
| Axillary | 31 (96.9) | 1 (3.1) | 1.00 | | | |
| Subclavian | 251 (96.2) | 10 (3.8) | 1.24 | 0.15 | 9.98 | |
| Jugular | 106 (94.6) | 6 (5.4) | 1.76 | 0.20 | 15.13 | |
| Atrium | 449 (98.2) | 8 (1.8) | 0.55 | 0.07 | 4.56 | |
| High pressure saline flush **successfully performed, n (%)** | | | | | | **0.030*** |
| No | 118 (93.7) | 8 (6.3) | 1.00 | | | |
| Yes | 266 (98.2) | 5 (1.8) | 0.28 | 0.09 | 0.87 | |
| **Tip confirmation method, n (%)** | | | | | | 0.079 |
| Radiography/Navigation/Scopy | 6464 (98) | 134 (2) | 1.00 | | | |
| ECG | 4897 (98.4) | 79 (1.6) | 0.78 | 0.59 | 1.03 | |
| Reversible catheter oclusion, **n (%)** | | | | | | **0.039** |
| No | 10391 (98.2) | 186 (1.8) | 1.00 | | | |
| Yes | 984 (97.3) | 27 (2.7) | 1.53 | 1.02 | 2.31 | |
| **Tip repositioning, n (%)** | | | | | | >0.999* |

(*Continued*)

**Table 4.** (Continued)

| Variable | Thrombosis (PICC) | | OR | 95% CI | | p |
|---|---|---|---|---|---|---|
| | No | Yes | | Inferior | Superior | |
| No | 11348 (98.2) | 213 (1.8) | 1.00 | | | |
| Yes | 27 (100) | 0 (0) | & | | | |

Chi-square test

* Fisher's exact test; # Likelihood ratio test

** Student's t-test; & Unable to estimate.

brazilian hospitals including 12,725 patients, catheter-related DVT rate was 1.0% and reversible catheter oclusion was 2.5% [13–16].

Regarding critically ill patients, a large study demonstrated a large variation in PICC use practices in ICU setting, DVT incidence could reach more than 10%, and a multivariate analysis showed that PICC in ICU had almost double risk of PICC-associated DVT, compared to ward patients. Our study did not find any independent further risk for DVT in critically ill patients. Nonetheless, further studies should confirm effectiveness and safety of PICC in ICU scenario. The present study adds to field in a way that we sought to identify, in a population with a high risk of thrombosis, which were the potential risk factors associated with PICC-associated DVT. For this purpose we were able to collect a vast array of variables related to either characteristics of the patients, devices and the insertion technique. Interestingly, the only independent variables associated to DVT were non-modifiable, intrinsic characteristics, such as age and history of previous DVT, and two potentially modifiable variables. The presence of obstruction associated with a higher risk, and anticoagulant use conferred protection. Both variables might be subjected to future interventions in order to decrease risk [17,18].

The use of CVC has become part of the routine management of hospitalized patients for chemotherapy, antibiotic therapy, and parenteral nutrition. In this cohort study, PICC

**Table 5. Adjusted model results to explain the occurrence of DVT due to PICC.**

| Variable | OR | 95% CI | | p |
|---|---|---|---|---|
| | | Inferior | Superior | |
| Sex (Women) | 1.18 | 0.89 | 1.57 | 0.241 |
| Age (years) | 1.011 | 1.002 | 1.020 | **0.013** |
| DVA | 1.36 | 0.94 | 1.96 | 0.099 |
| Chemotherapy | 1.53 | 0.94 | 2.50 | 0.090 |
| Oncological | 1.28 | 0.91 | 1.80 | 0.162 |
| Previous thrombosis | 1.96 | 1.12 | 3.44 | **0.018** |
| Anticoagulant | 0.73 | 0.54 | 0.99 | **0.040** |
| Punctured limb, n (%) | 0.85 | 0.64 | 1.13 | 0.261 |
| Venipuncture area | | | | |
| Green area | 1.00 | | | |
| Yellow area | 0.41 | 0.10 | 1.65 | 0.209 |
| Red area | & | | | 0.997 |
| Ideal zone | 0.89 | 0.61 | 1.30 | 0.553 |
| Tip confirmation method (ECG) | 0.80 | 0.59 | 1.08 | 0.148 |
| Obstruction | 1.60 | 1.05 | 2.42 | **0.028** |

Multiple logistic regression; & Unable to estimate.

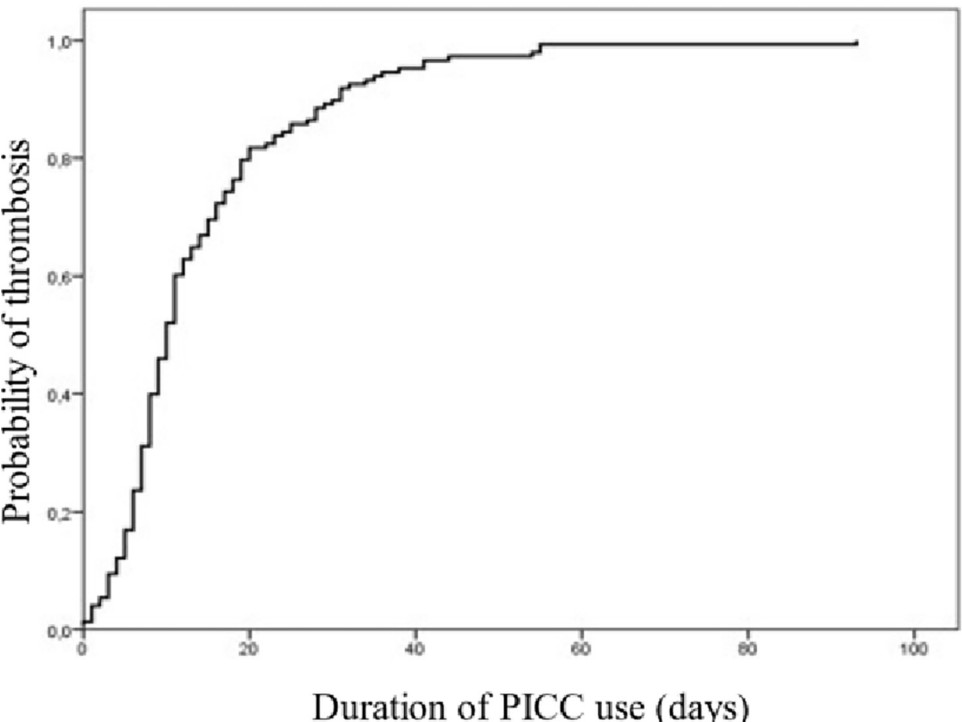

**Fig 2. DVT probability with the duration of PICC use (n = 213).**

insertion was exclusively performed by a team of dedicated nurses, with a success rate in the procedure between 95 to 99% and accomplished in the first attempt in 88.4%. Besides that, the adequate choice of the cathether size, the use of ultrasound guide and the help of devices that increase the chance of the proper location of the cathether tip might have contributed to the our results. Indeed, catheter-to-vein ratio in 91.5% procedures was < 33%, along with the majority being 5Fr or smaller, which must have contibuted to low rates of DVT [19].

As for the choice of limb, 51.2% of the procedures were in RUL. In a randomized clinical trial, the overall incidence of right-sided complications was 23% versus 34% on the left, confirming the hypothesis that right-sided insertions lead to fewer complications (p = 0.046) regardless of hand dominance, however, in another retrospective study the laterality of PICC insertion was not significantly associated with major complications [7,20,21].

In the present study, 61% of the procedures were performed in the basilic vein, and the site of punction did not associate with DVT risk. A retrospective study by Liem *et al*. demonstrated a DVT rate was 3.1% in the basilic vein and 2.2% in the brachial vein [22].

With regard to minor complication, catheter obstruction occurred in 8.7% and remained as an independent variable associated with PICC-related DVT. Obstruction is the most common PICC-related adverse event, and good device maintenance and management practices can reduce this adverse event. The PICC must be washed before and after the administration of medications using a pulsatile technique to reduce the risk of intraluminal occlusion, but there must have been other non modifiable factors related to patient [23].

In this study, the reason for the removal of PICC was the end of therapy in 67.9%, and a suspected CLABSI was noted in 14% of the cases. The implementation of protocols with a clinically valid indication for device insertion and removal must be implemented, avoiding overuse

in indication and mantainance. The low duration of the PICC in our cohort, based on the awareness for the prompt removal when possible, might have helped with our results.

The incidence of DVT, which is a threatening complication with this catheter, is current low, provided all the best technologies and practices in insertion and maintenance are employed. There has been a growth of PICC usage even in a middle-income countries as an effective alternative to other central venous lines, even in a scenario of high risk of DVT. We corroborate the increasing knowledge that even ICU and cancer patients, those with the highest risk for DVT might safely benefit with the use of PICC.

## Conclusion

This study indicates that PICC is a safe and suitable intravenous device for medium- and long-term intravenous therapy. Using PICC in a service with bundles and a specialized, fully trainned and dedicated vascular access nurse team evidenced a low incidence of DVT despite a high prevalence of patients with critical and oncological profiles in the sample.

## Study limitations

This study has some limitations. The first one is the single center and its retrospective aspect, although our sample is quite robust to allow interpretation and the collection of the data was performed prospectively. Secondly, we have an issue related to the reason for the use of anticoagulants. Our database did not allow us to separate between the use of prophylactic and therapeutic anticoagulants, although based on our hospital historical epidemiologic data, it is highly suggested that most of the anticoagulants were prophylactic (either low molecular weight or unfractioned heparin, based on a JCI approved institution-based DVT prophylaxis guideline), with a very low number of the patients receiving oral anticoagulants.

Another limitation is related to systemic infections or systemic inflammatory response syndrome evaluation regarding the risk of DVT, since most patients underwent the placement of the catheter for this reason, and we did not collect any biomarker for infection/inflammation that coud help predict DVT risk.

## Supporting information

**S1 File.**
(XLSX)

## Acknowledgments

Professor Eneida Rabelo Rejane da Silva, for her collaboration in this study.

## Author Contributions

**Conceptualization:** Telma Christina do Campo Silva, Jose Mauro Vieira Junior.

**Investigation:** Telma Christina do Campo Silva.

**Methodology:** Telma Christina do Campo Silva, Luciene Muniz Braga, Jose Mauro Vieira Junior.

**Resources:** Telma Christina do Campo Silva, Luciene Muniz Braga.

**Software:** Telma Christina do Campo Silva.

**Supervision:** Jose Mauro Vieira Junior.

**Validation:** Telma Christina do Campo Silva, Jose Mauro Vieira Junior.

**Visualization:** Jose Mauro Vieira Junior.

**Writing – original draft:** Telma Christina do Campo Silva, Jose Mauro Vieira Junior.

**Writing – review & editing:** Telma Christina do Campo Silva, Luciene Muniz Braga, Jose Mauro Vieira Junior.

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
