## [Decision Letter · Decision Letter 0]

20 Nov 2023

PONE-D-23-26732RISK OF DEEP VENOUS THROMBOSIS ASSOCIATED WITH PERIPHERALLY INSERTED CENTRAL CATHETER: A RETROSPECTIVE COHORT STUDY OF 11,588 CATHETERS IN BRAZILPLOS ONE

Dear Dr. MAZZETTO,

Thank you for submitting your manuscript to PLOS ONE. After careful consideration, we feel that it has merit but does not fully meet PLOS ONE’s publication criteria as it currently stands. Therefore, we invite you to submit a revised version of the manuscript that addresses the points raised during the review process.

We look forward to receiving your revised manuscript.

Kind regards,

Eyüp Serhat Çalık

Academic Editor

PLOS ONE

Journal Requirements:

5. Please ensure that you include a title page within your main document. You should list all authors and all affiliations as per our author instructions and clearly indicate the corresponding author.

6. Please amend either the abstract on the online submission form (via Edit Submission) or the abstract in the manuscript so that they are identical.

Additional Editor Comments:

I congratulate the authors for this valuable work.

The prevalence of DVT associated with PICC and its successful implementation deserves appreciation. The manuscript was reviewed by three peer reviewers, their comments are below. A recent meta-analysis (https://doi.org/10.1177/10760296221144041) found that the prevalence of PICC-related DVT increased significantly with increasing catheter diameters. Do you have a method to determine the size of catheter to be used? Do you determine the catheter size based on the patient's BMI or venous diameter? Please specify.

Reviewers' comments:

Reviewer's Responses to Questions

**Comments to the Author**

1. Is the manuscript technically sound, and do the data support the conclusions?

Reviewer #1: Yes

Reviewer #2: No

Reviewer #3: Yes

2. Has the statistical analysis been performed appropriately and rigorously? 

Reviewer #1: Yes

Reviewer #2: Yes

Reviewer #3: N/A

3. Have the authors made all data underlying the findings in their manuscript fully available?

Reviewer #1: Yes

Reviewer #2: Yes

Reviewer #3: Yes

4. Is the manuscript presented in an intelligible fashion and written in standard English?

Reviewer #1: Yes

Reviewer #2: No

Reviewer #3: Yes

5. Review Comments to the Author

Reviewer #1: Nice and sincere efforts spent to collet these large number of cases. The draft give important idea about PICC and DVT relation. Its considered as relevant and worth data that we could depend on its results.

Reviewer #2: The authors tried to demonstrate safety and feasibility of PICC therapy introduced by a specialized team. Although it might be an interesting report, there are several issues to be revised.

1) What is the main objective in this study? To analyze the risk factors for DVT or to check the competency of the special team? The conclusion does not correspond to the objective in the abstract and the manuscript as well.

2) Why could the authors conclude that “using PICC in a service with bundles and a specialized team evidenced a low incidence of DVT”? To which factors listed in the Table 3 and 4 did the team contribute or correlate?

3) There is little consistency and profound discussion in the Discussion section. Most paragraphs only demonstrate the data from the references. What are the new findings in this study? How do they differ from the previous studies? What is the clinical implication of the study?

4) There were too many errors of English grammar and word spelling. The Conclusion section should be written in English in the 1st page. The authors should get English proofreading again for the manuscript.

5) Moreover, the authors should check “Submission Guidelines” of this Journal again. The title page is necessary and References should be listed according to the PLOSONE guidelines.

Reviewer #3: I think the analysis results based on the large number of cases are excellent in this paper. This paper is worthy of publication. I would like to ask you a few questions below.

1. What kind of treatment is positive pressure (flush) successfully performed?

2. Depending on the guidelines, patients with a history of thrombosis may be given anticoagulants, which may be beneficial in preventing thrombosis. How often do they experience bleeding?

3. If PICC is indicated for a patient with cancer and a history of thrombosis, anticoagulants will eventually be indicated. It was important to take into account which anticoagulants were indicated or PICC were introduced, these treatments were depended on the degree of progression of the cancer. Medical systems differ in many countries, which do you prefer anticoagulants or use of PICC in your country?

4. This underlined part, death 11.4%) in the results of abstract, is incorrect.

6. PLOS authors have the option to publish the peer review history of their article (what does this mean?). If published, this will include your full peer review and any attached files.

Reviewer #1: **Yes: **Aram Baram

Reviewer #2: **Yes: **Hiroki Hata

Reviewer #3: No

---

## [Author Response · Author response to Decision Letter 0]

18 Feb 2024

Response to Editor and reviewers:

I congratulate the authors for this valuable work. The prevalence of DVT associated with PICC and its successful implementation deserves appreciation.

We deeply thank to the Editor for the appreciation of our study and the opportunity given in order to improve the manuscript. You will be able to see that we made modifications based on the commentaries/suggestions received, and we reviewed the format and references altogether.

 The manuscript was reviewed by three peer reviewers, their comments are below. A recent meta-analysis (https://doi.org/10.1177/10760296221144041) found that the prevalence of PICC-related DVT increased significantly with increasing catheter diameters.

Thank you for the commentaries. We have now added this recent, important paper to the references list. And we incorporated to the manuscript (results and discussion), the aspect of the size of the device. We made it clearer that most of our PICC inserted were 5F or smaller, which might have contributed to our good results.

Do you have a method to determine the size of catheter to be used? Do you determine the catheter size based on the patient's BMI or venous diameter? Please specify.

We appreciate your comments. Yes, we do have a method for determining the size of vein, based on ultrasound measurements, and therefore catheter to be used. First, we define the number of lumen`s catheter according to the patient’s diagnosis, comorbidities, intravenous therapy and time of hospital stay. And employing an ultrasound we then estimate vein diameter in order to choose a cathether that would end up respecting the suggested ratio (The current Infusion Nurses Society guidelines 2021 recommend the use of PICC`s with a catheter-to-vein ratio of no more than 45% prior to insertion of a vascular access device in the upper extremity, and we follow this guideline).

More specifically, during the data collection period, two equipments were used. The Site Rite 5 (BD), was used from February 2012 to January 2017, which gives us a more limited view. Thereafter, the site rite 8 (BD) started to be used from February 2017 until nowadays. This equipment provides a more accurate catheter-to-vein ratio, in three-dimensional way. We do not consider BMI in our decision-making process.

Reviewer #1: Nice and sincere efforts spent to collet these large number of cases. The draft give important idea about PICC and DVT relation. Its considered as relevant and worth data that we could depend on its results.

We appreciate your comments. We used the RedCap to collect the number of cases. We did this in a retrospective way and now we made it clearer in the manuscript.

Reviewer #2: The authors tried to demonstrate safety and feasibility of PICC therapy introduced by a specialized team. Although it might be an interesting report, there are several issues to be revised.

We are deeply grateful for your insightful commentaries and suggestions, that will definitely improve our manuscript. We hope we now comply to your requests (see below).

1) What is the main objective in this study? To analyze the risk factors for DVT or to check the competency of the special team? The conclusion does not correspond to the objective in the abstract and the manuscript as well.

Thanks for the precise notation. Our main objective is to describe the actual rate and risk factors for DVT in PICC patients in a cohort of a tertiary hospital in a Middle income country, particularly in scenario of cancer and critically-ill patients, which we understand could contribute and predispose to this complication. We hope we now made it more clear in the current version of the manuscript. Nevertheless, since these data we obtained might not necessarily apply to other Middle incomes countries services, we tried to ensure that those data might have resulted from some features. For instance, our hospital is accredited by Joint Commission International (JCI) and we have a service with bundles and a specialized nurses fully trainned and dedicated to the purpose of the PICC insertion and maintainnance, which may not be the case in other services. Yet, we appreciate the comment about conclusion, and we have rewritten this topic.

2) Why could the authors conclude that “using PICC in a service with bundles and a specialized team evidenced a low incidence of DVT”? To which factors listed in the Table 3 and 4 did the team contribute or correlate?

We figured that if the technical aspects of insertion had varied regarding choice of the site, type and syze of the catheter, catheter-to-vein ratio and tip location, those could have been independent contributing variables to DVT. However, data in table ensure that the there was an overall compliance to best practices. We now state in the manuscript that inference. Yet, the team had 88.4% the puncture in the first attemp, 91,5% occupancy < 33% and 98,3% of procedures in the ideal zone, 98.1% tip location in the superior vena cava. Data could have been different in case of a distinct organization and with inferior quality indicators. 

3) There is little consistency and profound discussion in the Discussion section. Most paragraphs only demonstrate the data from the references. What are the new findings in this study? How do they differ from the previous studies? What is the clinical implication of the study?

We thank you for the comments about the topic discussion. As we can see in the new discussion section, we sought to follow the topics we pointed. In summary: our study adds to field in a way that we sought to identify, in a population with a high risk of thrombosis, which were the potential risk factors for PICC-associated DVT. The incidence of DVT, which is a threatening complication with this catheter, is current low, provided all the best technologies and practices in insertion and maintenance are employed. And yet, the study shows a regression analysis finding independent variables associated with the DVT risk, which might shed light to future studies. We corroborate to the increasing knowledge that even ICU and cancer patients, those with the highest risk for DVT might safely benefit with the use of PICC. We included some data on ICU patients from the literature.

4) There were too many errors of English grammar and word spelling. The Conclusion section should be written in English in the 1st page. The authors should get English proofreading again for the manuscript.

We apologize for the typos and errors of English grammar. We have now thoroughfully revised the manuscript.

5) Moreover, the authors should check “Submission Guidelines” of this Journal again. The title page is necessary and References should be listed according to the PLOSONE guidelines. Reviewer #2: The authors tried to demonstrate safety and feasibility of PICC therapy introduced by a specialized team. Although it might be an interesting report, there are several issues to be revised.

We attached the tittle page, and we adjusted the references according to the PLOS ONE.

Reviewer #3: I think the analysis results based on the large number of cases are excellent in this paper. This paper is worthy of publication. I would like to ask you a few questions below.

1) What kind of treatment is positive pressure (flush) successfully performed?

We appreciate your relevant questions. We use positive pressure in polyurethane catheters. This catheter has greater rigidity, low thrombogenicity and thermosensitive, which allows us sometimes to reposition the device. For instance, when we observed after insertion that it was positioned in the subclavian vein, and even when it migrated to an innapropriate position, in these cases we use/apply positive pressure with saline solution to reposition the catheter in the vena cava superior. 

2) Depending on the guidelines, patients with a history of thrombosis may be given anticoagulants, which may be beneficial in preventing thrombosis. How often do they experiencie bleeding?

That is an interesting question. Unfortunately our database did not allow us to separate between the use of prophylactic and therapeutic anticoagulants, although based on our hospital historical epidemiologic data, it is highly suggested that most of the anticoagulants in our cohort were prophylactic (we now made it more clear in the manuscript and table). Besides that, the kit to insert the guidewire and thereafter the catheter is based on a small needle micropuncture, and bleeding is not an expected complication. Last, bleedings related to the insertion on maintenance of a PICC device would be reported as an adverse event in our electronic medical record-based reporting database, and we could not find adverse events related to bleeding, though.

3) If PICC is indicated for a patient with cancer and a history of thrombosis, anticoagulants will eventually be indicated. It was important to take into account which anticoagulants were indicated or PICC were introduced, these treatments were depended on the degree of progression of the cancer. Medical systems differ in many countries, which do you prefer anticoagulants or use of PICC in your country?

Our database did not allow us to separate between the use of prophylactic and therapeutic anticoagulants, although based on our hospital historical epidemiologic data, it is highly suggested that most of the anticoagulants were prophylactic (either low molecular weight or unfractioned heparin) based on a JCI approved institution-based DVT prophylaxis guideline, with a very low number of the patients receiving oral anticoagulants. As mentioned above, we now made it more clear in the manuscript and we now depicted these data in table (we now can see clearly that most of the patients were under low weight or unfractioned heparin, with smaller portion of the cohort receiving either warfarin or new antithrombotic drugs).

4) This underlined part, death 11.4% in the results of abstract, is incorrect.

Thank you for the important observation. It was a lapse, and now we correctly mention the mortality rate of the cohort in the proper place in the manuscript. As we can also see, the abstract was rewritten not to adjust for the lenght, but also to comply with the most important informations of the study.

---

## [Editor Report · Decision Letter 1]

28 Feb 2024

Risk of Deep Venous Thrombosis associated with Peripherally Inserted Central Catheter: a retrospective cohort study of 11.588 catheters in Brazil

PONE-D-23-26732R1

Dear Dr. Mazzetto,

We’re pleased to inform you that your manuscript has been judged scientifically suitable for publication and will be formally accepted for publication once it meets all outstanding technical requirements.

Kind regards,

Eyüp Serhat Çalık

Academic Editor

PLOS ONE